# *TGFB2* Gene Methylation in Tumors with Low CD8^+^ T-Cell Infiltration Drives Positive Prognostic Overall Survival Responses in Pancreatic Ductal Adenocarcinoma

**DOI:** 10.3390/ijms26125567

**Published:** 2025-06-10

**Authors:** Vuong Trieu, Michael Potts, Scott Myers, Stephen Richardson, Sanjive Qazi

**Affiliations:** 1Oncotelic Therapeutics, 29397 Agoura Road, Suite 107, Agoura Hills, CA 91301, USA; vtrieu@oncotelic.com (V.T.); michael.potts@oncotelic.com (M.P.); scott.myers@oncotelic.com (S.M.); stephen.richardson@oncotelic.com (S.R.); 2Westmorland Campus, Kendal College, Market Place, Kendal, Cumbria LA9 4TN, UK

**Keywords:** biomarker, adenocarcinoma, prognosis, transforming growth factors, tumor microenvironment, immunotherapy, CD8^+^, T-cell, methylation, interferon-alpha

## Abstract

Pancreatic ductal adenocarcinoma (PDAC) typically exhibits asymptomatic clinical features, with most patients diagnosed at an advanced metastatic stage. Current treatment options are limited to cytotoxic standard therapies, primarily FOLFIRINOX or modified FOLFIRINOX regimens. This highlights a critical need for targeted therapies to improve efficacy and reduce toxicity. We have sought to identify potential biomarkers based on DNA methylation profiles to identify patient groupings with improved overall survival (OS) based on the Transforming Growth Factor Beta (TGFB) gene complex, and the interferon-related pathway gene, *IFI27*, using the TCGA dataset for PDAC patients. We employed a multivariate Cox proportional hazards model to directly compare hazard ratios for *TGFB1/2/3* and *IFI27* methylation impacting OS. We also controlled for age at diagnosis, sex, and *TGFB2* gene methylation by examining the statistical interactions between the marker gene mRNA expression and the TGFB2 gene. Genes were filtered based on the tumor-specific expression patterns and Cox models with highly significant interaction terms to identify mRNA expression of genes that amplified the impact of *TGFB2* methylation. The effect of the *TGFB2* gene methylation in the context of marker gene mRNA expression was analyzed using Kaplan–Meier (KM) analysis. Marker genes were correlated to T-cell enrichment patterns using the deconvolution algorithms provided by the TIMER 2.0 database. Methylation of *TGFB2*, *TGFB3* and *IFI27* genes using median cut-off values for KM plots showed significant improvements in median overall survival of 5.7 (*p* = 0.044), 5.2 (*p* = 0.036), and 3.7 (*p* = 0.028) months for high methylation levels for *TGFB2*, *IFI27*, and *TGFB3* genes, respectively. In contrast, high levels of *TGFB1* methylation exhibited a shorter 4.7 (*p* = 0.016) month median OS time. The impact of *TGFB2* methylation was amplified at low expressions of marker genes that were highly correlated with CD8^+^ T-cell infiltration. Patients with high levels of *TGFB2* methylation when compared to low levels of *TGFB2* methylation showed median overall survival (OS) improvements at low mRNA expression levels: 54.2 months for CD3D (*p* < 0.0001); 54 months for LCK (*p* = 0.0009); 54.9 months for HLA-DRA (*p* = 0.0001); and 9 months for RAC2 mRNA expression (*p* = 0.0057). *TGFB2* gene methylation drives TGFB2 mRNA expression to achieve clinical impact, as high levels of TGFB2 mRNA, at low levels of the marker genes, resulted in worse median OS times. *TGFB2* methylation is a prognostic marker for PDAC patients within an immunosuppressed tumor microenvironment characterized by low CD8^+^ T-cell infiltration. This correlation is functionally associated with TGFB2 mRNA production, suggesting that targeting TGFB2 mRNA through knockdown can potentially enhance PDAC prognosis.

## 1. Introduction

Pancreatic cancer accounts for almost as many deaths (466,000) as cases (496,000) because of its poor prognosis; it is the seventh leading cause of cancer death in both sexes [1]. The mortality rate has gradually increased by 0.3% annually among men since the year 2000, while among women, this trend has persisted since at least 1975, reflecting analogous patterns in incidence [2]. Its high fatality rate is due in part to asymptomatic clinicopathology, with 80% of patients presenting with metastases at the time of diagnosis [3]. Surgical resection remains the sole curative intervention for early-stage local disease affecting 20–30% of patients [4]. However, tumor recurrence in pancreatic cancer occurs in up to 85% of these individuals [5]. Furthermore, approximately 70–80% of patients do not benefit from surgical intervention, as they present with advanced disease and metastasis at the time of diagnosis [6]. For these patients, cytotoxic chemotherapy has become the standard of care, employing either nab-paclitaxel and Gemcitabine or a combination of folinic acid, fluorouracil, leucovorin, irinotecan, and oxaliplatin (FOLFIRINOX or modified FOLFIRINOX), extending median overall survival to approximately 14 months [7,8,9,10,11,12,13]. Thus, an unmet need exists for innovative targeted therapies that exhibit reduced toxicity and increased efficacy. Presently, numerous clinical trials are underway to assess the efficacy of inhibitors targeting druggable mutations, including KRAS, PARP, and SHP2 [14,15,16]. Clinical trials investigating the efficacy of immune checkpoint monotherapies in PDAC have demonstrated disappointingly low response rates [17,18,19].

Biomarker-led stratification of patients demonstrating a favorable prognosis holds significant potential to enhance pancreatic cancer treatment by enabling more precise and less toxic therapeutic strategies compared to the current standard FOLFIRINOX-based therapies. For example, hENT1/hCNT3 expression identifies patients likely to benefit from Gemcitabine, with high-expression patients showing a 54.6% three-year survival rate versus 26.1% in low-expression groups [20]. A recent study highlighted the potential role of Ubiquitin Specific Peptidase 39 (USP39) as a valuable immune-related biomarker with both diagnostic and prognostic utility across multiple cancer types, especially PDAC, emphasizing its promise as a therapeutic target for cancer immunotherapy [21].

We have sought to identify potential biomarkers based on DNA methylation profiles in patients with pancreatic ductal adenocarcinoma (PDAC). DNA methylation offers significant advantages over mRNA levels for biomarker discovery due to its chemical stability, tissue specificity, and direct link to gene regulation, providing functional insights and the potential for early detection. Technologies, such as bisulfite sequencing, enable precise assessments of DNA methylation patterns, making them highly effective for early diagnosis and clinical decision-making [22]. DNA methylation biomarkers consistently demonstrate higher accuracy in cancer detection and classification compared to mRNA expression profiles. Recent comparative analyses show that DNA methylation-based classifiers significantly outperform mRNA-based approaches in tumor tissue origin detection. Studies utilizing The Cancer Genome Atlas (TCGA) data reveal that DNA methylation-based models achieve an impressive overall accuracy of 97.77%, substantially exceeding the 88.01% accuracy of mRNA expression models [23].

Gene methylation plays a significant role in the prognosis of pancreatic cancer, serving as a biomarker for both survival and treatment responses. One study identified a large repertoire of 1235 differentially methylated DNA genes comparing PDAC and adjacent tissues, with 78 methylation markers identified as independently affecting PDAC prognosis [24]. This research highlighted the potential for DNA methylation profiles to reveal tumor heterogeneity and inform treatment strategies for subsets of PDAC patients based on prognostic relevance. Another study developed a tissue-based DNA methylation risk-score model that predicted overall survival in surgically resected pancreatic cancer patients. This model correlated high-risk methylation signatures with significantly poorer survival outcomes, indicating that such models could aid in clinical decision-making [25]. Moreover, research has shown that specific genes, like SPARC and NPTX2, exhibit elevated methylation levels in pancreatic cancer patients; their methylation status correlates with advanced disease stages and poor survival [26]. Similarly, a novel four-DNA methylation model was found to predict prognosis in PDAC patients, demonstrating high sensitivity and specificity in survival predictions [27]. In summary, gene methylation is increasingly recognized as a potentially useful factor in the prognosis of PDAC, providing insights into tumor biology, aiding in risk stratification, and offering potential targets for therapeutic intervention [28,29].

Recently, we characterized the positive prognostic value of methylation of the Transforming Growth Factor Beta 2 (*TGFB2*) gene (hazard ratio (HR) (95% CI range) = 0.04 (0.006–0.274); *p* = 0.001), which was greater than the well-established impact of the O-6-Methylguanine-DNA Methyltransferase (*MGMT*) gene methylation (HR (95% CI range) = 0.667 (0.475–0.936); *p* = 0.019) interrogating the TCGA dataset for glioblastoma patients [30]. Furthermore, *TGFB2* gene methylation was negatively correlated with Reactome pathways representing T-cell activation and effector functions [30]. T-cells are also instrumental to the immune response against PDAC, with their functionality substantially shaped by TGFB ligands. TGFB facilitates the conversion of naive CD4^+^ T cells into CD4^+^Foxp3^+^ regulatory T-cells (Tregs), which inhibit anti-tumor immunity, thereby promoting tumor progression [31]. Within the PDAC tumor microenvironment (TME), elevated TGFB levels contribute to fibrosis and immune evasion, thereby enhancing tumorigenesis and suppressing cytotoxic T-cell activity [32]. Initially acting as a tumor suppressor by inhibiting epithelial cell growth, TGFB can shift its role in advanced stages of PDAC to promote tumor progression through augmented Treg functionality and diminished cytotoxic T-cell responses [33]. CD8^+^ T-cells, or cytotoxic T lymphocytes, are also integral to the immune response against PDAC. However, the immunosuppressive tumor microenvironment mitigates their anti-tumor impact. In PDAC, CD8^+^ T-cells often display elevated levels of PD-1 expression, which correlates with diminished activation and cytotoxic activity, while tumor cells express PD-L1, which further inhibits CD8^+^ T-cell activation [34]. TGFB additionally contributes to shaping the TME by promoting M2 macrophages and releasing interleukin-10 (IL-10), leading to an immunosuppressive cytokine milieu that contributes to a decreased immune response, ultimately hindering the effective action of CD8^+^ T-cells against tumor cells [35,36]. Notably, while depleting Tregs can modify the tumor microenvironment, it may paradoxically accelerate tumor progression instead of enhancing CD8^+^ T-cell effectiveness [37], suggesting a complex interplay between different subsets of T-cells in the TME. To address these challenges, various therapeutic strategies are being investigated, including immune checkpoint inhibitors that target PD-1 and CTLA-4, combined treatment regimens to enhance tumor immunogenicity, and innovative methods aimed at depleting immunosuppressive factors within the tumor milieu [38]. Thus, while CD8^+^ T-cells are essential for anti-tumor immunity in pancreatic cancer, their functional capacity is significantly hampered by the tumor’s immunosuppressive environment, thereby influencing patient outcomes.

Our previous study demonstrated that activation of the interferon-alpha pathway and elevated TGFB2 mRNA levels are associated with worse overall survival in PDAC patients [39]. In this research, we discovered that elevated mRNA expression levels have significant prognostic implications. Patients exhibiting high expression of either TGFB2, IRF9, or IFI27 mRNA had median overall survival (mOS) times between 16 and 20 months. This is in contrast to those with low expression levels of both TGFB2 and either IRF9 or IFI27 mRNA, who demonstrated a mOS of 72 months, with a statistically significant difference (*p* < 0.01) [39]. Given the potential for using DNA methylation as prognostic biomarkers, we have extended these studies to use a bioinformatic-driven approach to characterize the impact of *TGFB1/2/3* and *IFI27* gene methylation on the overall survival in PDAC patients. We implemented a multivariate Cox proportional hazards model to directly compare hazard ratio (HR) calculations for *TGFB1/2/3* and *IFI27* gene methylations on OS, controlling for age at diagnosis and sex variables. Genome-wide screening of mRNA expression values correlated to Reactome pathways identified signaling cascades activated by T-cell engagement and negatively correlated with *TGFB2* gene methylation. A second Cox proportional hazards model, which included marker gene mRNA expression levels and quantified the marker gene by the *TGFB2* gene methylation interaction term, identified four marker genes whose expression was highly correlated to CD8^+^ T-cell infiltration (CD3D, LCK, HLA-DRA, and RAC2). At low marker gene expression levels, high levels of *TGFB2* gene methylation and low levels of TGFB2 mRNA resulted in significant positive impacts on PDAC OS. These observations suggest that *TGFB2* methylation can be used as a prognostic marker in PDAC patients in immunosuppressed TME, and that this is functionally linked to TGFB2 mRNA production, thereby also providing a target for therapy, as knockdown of TGFB2 mRNA is hypothesized to improve PDAC prognosis.

## 2. Results

### 2.1. Positive Prognostic Impact of High TGFB2/3 and IFI27, and Negative Prognostic Impact of TGFB1 Gene Methylation Levels on Overall Survival (OS) in Adult PDAC Patients

We compared PDAC patients with high versus low gene methylation levels stratified using median cut-offs using Kaplan–Meier analysis. The *TGFB2^highMe^* group of patients (median overall survival (mOS) = 22.7; 95% CI = 19.8–NA months; *n* = 89; #death events = 43) exhibited a significantly longer OS outcome than the *TGFB2^lowMe^* group of patients (median = 17.0; 95% CI = 15.3–21.9 months; *n* = 89; #death events = 50; log-rank chi-square = 4.06, *p* = 0.044) (Figure 1A). The *IFI27^highMe^* group of patients (mOS = 22.7; 95% CI = 19.6–NA months; *n* = 89; #death events = 41) exhibited a significantly longer OS outcome than *IFI27^lowMe^* patients (mOS = 17.5; 95% CI = 15.3–23.08 months; *n* = 89; #death events = 52; log-rank chi-square = 4.39, *p* = 0.036) (Figure 1B). The *TGFB1^highMe^* group of patients (mOS = 19.4; 95% CI = 15.8–21.7 months; *n* = 89; #death events = 55) exhibited a significantly shorter OS outcome than the *TGFB1^lowMe^* group of patients (mOS = 24.1; 95% CI = 18.7–NA months; *n* = 89; #death events = 38; log-rank chi-square = 5.84, *p* = 0.016) (Figure 1C). The *TGFB3^highMe^* group of patients (mOS = 23.1; 95% CI = 18.7–NA months; n = 89; #death events = 40) exhibited a significantly longer OS outcome than the *TGFB3^lowMe^* group of patients (mOS = 19.4; 95% CI = 15.1–21.4 months; *n* = 89; #death events = 53; log-rank chi-square = 4.86, *p* = 0.028) (Figure 1D).

To assess whether gene methylation can functionally drive the mRNA product, resulting in a prognostic impact in PDAC patients, we first correlated gene methylation with the corresponding mRNA (Appendix A). Then, we evaluated the prognostic impact of mRNA levels on overall survival (OS) (Appendix A). More than 30% of the variation was explained for the positive correlation of *TGFB2* (R^2^ = 0.316, *p* < 0.001), *IFI27* (R^2^ = 0.356, *p* < 0.001), and *TGFB3* (R^2^ = 0.322, *p* < 0.001) methylation with the corresponding mRNA. *TGFB1* gene methylation exhibited a weak correlation with TGFB1 mRNA (R^2^ = 0.05, *p* = 0.003). These results suggest that *TGFB2*, *IFI27*, and *TGFB3* gene methylations can drive mRNA products for the corresponding genes (Appendix A). The correlations between *TGFB2* and *IFI27* gene methylation and mRNA expression were prognostically associated with OS outcomes, comparing high versus low expression. This was not observed for the correlation between TGFB1 and TGFB3 mRNA and median OS outcomes (mOS) (Appendix A). The TGFB2^high^ subset of patients (mOS = 19.4; 95% CI = 15.34–22.71 months; *n* = 89; # death events = 51) exhibited a significantly shorter OS outcome than TGFB2^low^ patients (mOS = 21.71; 95% CI = 18.66–NA months; *n* = 88; #death events = 41, *p* = 0.034) (Appendix A). The IFI27^high^ patients (mOS = 16.8; 95% CI = 15.12–22.47 months; *n* = 89; #death events = 52) exhibited a significantly shorter OS outcome than IFI27^low^ patients (mOS = 24.05; 95% CI = 19.65–NA months; *n* = 88; #Events = 40; *p* = 0.009) (Appendix A).

### 2.2. TGFB1/2/3, and IFI27 Gene Methylation OS Impacts on PDAC Patients, Controlling for Age and Sex

We used the multivariate model to determine the impact of gene methylations on overall survival (OS) as independent variables, which considers correlations between the variables in the calculations for hazard ratios. There was a favorable prognostic impact, indicated by a hazard ratio (HR) less than one, for the *TGFB2^highMe^* group of patients (HR (95% CI range) = 0.53 (0.334–0.843); *p* = 0.007); the *TGFB3^highMe^* group of patients (HR (95% CI range) = 0.657 (0.43–1.002); *p* = 0.051); and the *IFI27^highMe^* group of patients (HR (95% CI range) = 0.473 (0.302–0.74); *p* = 0.001). The *TGFB1^highMe^* group of patients predicted worse OS outcomes (HR (95% CI range) = 1.631 (1.069–2.488); *p* = 0.023). These predicted hazard ratios for gene methylations were observed controlling for both the age at diagnosis (HR (95% CI range) = 1.03 (1.009–1.052); *p* = 0.006) and sex (HR (95% CI range) = 0.86 (0.569–1.3); *p* = 0.475) of the patients (Figure 2).

### 2.3. Identification of Prognostic Marker Genes from Enriched Reactome Pathways Correlating mRNA Levels with TGFB1/2/3 and IFI27 Gene Methylations

Beta values for *TGFB1/2/3* and *IFI27* gene methylation were correlated using Spearman ranks with mRNA expression levels of 14,861 genes for all PDAC patients (*n* = 177 evaluable patients) across 1286 Reactome pathways. We applied a *p*-value filter (*p* <0.0001) to identify highly correlated pathways to gene methylations. This identified 11 pathways negatively correlated with *TGFB2/3* and *IFI27* gene methylations and positively correlated with *TGFB1* gene methylation (Appendix A). Next, we identified 39 pathways negatively correlated with *IFI27* gene methylations (Appendix A). We also identified 27 *TGFB2* gene methylation-specific pathways using the following filter: *p*-value *TGFB2* methylation < 0.05 & NES *TGFB2* methylation < 0 & *p*-value *TGFB1* methylation > 0.1 & *p*-value *TGFB3* methylation > 0.1 & *p*-value *IFI27* methylation > 0.1. This filter identified 358 genes representing these 27 Reactome pathways (Appendix A). We compiled 856 genes by combining the gene lists from Appendix A that identified 358 genes negatively correlated (*p* < 0.01, FDR = 0.06) to *TGFB2* or *IFI27* gene methylations. Of these 356 genes, 41 genes exhibited a greater than 20-fold increase in expression (*p* < 0.0001, FDR < 0.0001) (Figure 3, Appendix A). We further processed these upregulated genes to determine the prognostic impact utilizing the multivariate Cox proportional hazards model, which identified 14 marker genes that statistically interacted with *TGFB2* gene methylation, affecting overall survival in PDAC patients (Appendix A). Four marker genes with a statistically significant interaction term were further investigated using Kaplan–Meier analysis.

### 2.4. Low Levels of Marker Gene mRNA Expression Amplify the Positive Prognostic OS Impact of TGFB2 Gene Methylation

The mOS for 30 patients from the CD3D^low^/*TGFB2^lowMe^* group was 12.7 (95% CI: 9.1–15.4, #death events = 24) months, while for 58 patients in the CD3D^low^/*TGFB2^highMe^* group, it was 66.9 (95% CI: 21.71–NA, #death events = 24) months. In the CD3D^high^/*TGFB2^lowMe^* group, the mOS for 59 patients was 21.9 (95% CI: 17.0–NA, #death events = 26) months; for 30 patients from the group CD3D^high^/*TGFB2^highMe^*, it was 19.8 months (95% CI: 15.3–NA, #death events = 18). At low levels of CD3D mRNA expression, increasing *TGFB2* gene methylation resulted in the most significant increase in OS times; there was a significant difference in OS outcomes comparing the CD3D^low^/*TGFB2^lowMe^* (*n* = 30, mOS = 12.7) versus the CD3D^low^/*TGFB2^highMe^* (*n* = 58, mOS = 66.9 months, OS difference = 54.2, *p* < 0.0001) groups of patients. Favorable OS outcomes were observed at increasing CD3D mRNA expression at low levels of *TGFB2* gene methylation; there was a significant difference in OS outcomes comparing the CD3D^low^/*TGFB2^lowMe^* (*n* = 30, mOS = 12.7 months) versus the CD3D^high^/*TGFB2^lowMe^* (*n* = 59, mOS = 21.9 months, OS difference = 9.2, *p* = 0.0002) groups of patients. However, a switch in the prognostic impact of CD3D occurred at high levels of *TGFB2* gene methylation, whereby there was a significant difference in OS outcomes comparing the CD3D^low^/*TGFB2^highMe^* (*n* = 58, mOS = 66.9 months) versus the CD3D^high^/*TGFB2^highMe^* (*n* = 30, mOS = 19.8 months, OS difference = −47.1, *p* = 0.04) groups of patients (Figure 4A).

The mOSs for four patient groups were as follows: for 35 patients in the LCK^low^/*TGFB2^lowMe^* group, the mOS was 12.9 months (95% CI: 9.1–20.2, #death events = 25); for 53 patients in the group LCK^low^/*TGFB2^highMe^*, it was 66.9 months (95% CI: 20.8–NA, #death events = 21); for 54 patients in the LCK^high^/*TGFB2^lowMe^* group, it was 21.9 months (95% CI: 16.4–NA, #death events = 25); and for 35 patients in the group LCK^high^/*TGFB2^highMe^*, the mOS was 19.5 months (95% CI: 15.5–NA, #death events = 21). At low levels of LCK mRNA expression, increasing *TGFB2* gene methylation resulted in the most significant increase in OS times; there was a significant difference in OS outcomes comparing the LCK^low^/*TGFB2^lowMe^
*(*n* = 35, mOS = 12.9 months) versus the LCK^low^/*TGFB2^highMe^* (*n* = 53, mOS = 66.9 months, OS difference = 54, *p* = 0.0009) groups of patients. At low levels of *TGFB2* gene methylation, a positive prognostic impact was observed at increasing LCK mRNA expression; there was a significant difference in OS outcomes comparing the LCK^low^/*TGFB2^lowMe^* (*n* = 35, mOS = 12.9 months) versus the LCK^high^/*TGFB2^lowMe^* (*n* = 54, mOS = 21.9 months, OS difference = 9, *p* = 0.0057) groups of patients (Figure 4B).

The mOS for 29 patients from the HLA-DRA^low^/*TGFB2^lowMe^* group was 12.0 (95% CI: 9.1–15.1, #death events = 21) months, while for 59 patients in the HLA-DRA^low^/*TGFB2^highMe^* group, it was 66.9 (95% CI: 21.71–NA, #death events = 24) months. For 60 patients from the HLA-DRA^high^/*TGFB2^lowMe^* group, the mOS was 20.2 (95% CI: 17.9–NA, #death events = 29) months; for 29 patients in the HLA-DRA^high^/*TGFB2^highMe^* group, it was 19.5 (95% CI: 12.3–NA, #death events = 18) months. At low levels of HLA-DRA mRNA expression, increasing *TGFB2* gene methylation resulted in a more favorable mOS outcome; there was a significant difference in mOS outcomes comparing the HLA-DRA^low^/*TGFB2^lowMe^* (*n* = 29, mOS = 12.0 months) versus the HLA-DRA^low^/*TGFB2^highMe^* (*n* = 59, mOS = 66.9 months, OS difference = 54.9, *p* = 0.0001) groups of patients. At low levels of *TGFB2* methylation, increasing HLA-DRA mRNA expression resulted in a more favorable OS outcome in PDAC patients; there was a significant difference in mOS outcomes comparing the HLA-DRA^low^/*TGFB2^lowMe^* (*n* = 29, mOS = 12.0 months) versus the HLA-DRA^high^/*TGFB2^lowMe^* (*n* = 60, mOS = 20.2 months, OS difference = 8.2, *p* = 0.0017) groups of patients. In contrast, at high levels of *TGFB2* gene methylation, an increase in HLA-DRA mRNA expression resulted in a worse prognosis; there was a significant difference in mOS outcomes comparing the HLA-DRA^low^/*TGFB2^highMe^* (*n* = 59, mOS = 66.9 months) versus the HLA-DRA^high^/*TGFB2^highMe^* (*n* = 29, mOS = 19.5 months, OS difference = −47.4, *p* = 0.017) groups of patients (Figure 4C).

The mOS for 41 patients from the RAC2^low^/*TGFB2^lowMe^* group was 15.1 (95% CI: 12.4–21.4, #death events = 26) months. The mOS for 47 patients in the RAC2^low^/*TGFB2^highMe^* group was 24.1 (95% CI: 20.8–NA, #death events = 20) months. In the RAC2^high^/*TGFB2^lowMe^* group, 48 patients had an mOS of 19.7 (95% CI: 17.0–NA, #death events = 24) months. Lastly, the mOS time for the 41 patients in the RAC2^high^/*TGFB2^highMe^* group was 17.5 (95% CI: 12.9–NA, #death events = 22) months. At low levels of RAC2 mRNA expression, increasing *TGFB2* gene methylation resulted in a more favorable mOS outcome; there was a significant difference in mOS outcomes comparing the RAC2^low^/*TGFB2^lowMe^* (*n* = 41, mOS = 15.1 months) versus the RAC2^low^/*TGFB2^highMe^* (*n* = 47, mOS = 24.1 months, OS difference = 9, *p* = 0.0057) groups of patients (Figure 4D).

### 2.5. Low Levels of Marker Gene mRNA Expression Amplify the Negative Prognostic OS Impact of TGFB2 mRNA Expression

Weak correlations were observed for CD3D (< 10% of the explained variation; R^2^ = 0.042, *p* = 0.007), LCK mRNA (R^2^ = 0.021, *p* = 0.054), and RAC2 (R^2^ = 0.079, *p* < 0.001) with TGFB2 mRNA. Correlation of HLA-DRA with TGFB2 mRNA explained greater than 10% of the variation (R^2^ = 0.149, *p* < 0.001) (Appendix A). These observations suggested that the stratification of patients according to TGFB2 and marker gene expression would result in relatively independent cohorts of four patient groups for OS comparisons.

To test whether the *TGFB2* gene methylation drives the prognostic impact of TGFB2 mRNA, we also compared high versus low levels of TGFB2 mRNA paired with the four marker genes (Figure 5). The mOS for 48 CD3D^low^/TGFB2^low^ mRNA group patients was 66.9 months (95% CI: 17.5–NA, #death events = 21). The mOS for 40 patients in the CD3D^low^/TGFB2^high^ mRNA group was 15.1 months (95% CI: 10.1–23.1, #death events = 27). In the CD3D^high^/TGFB2^low^ mRNA group of 40 patients, the mOS was 19.8 months (95% CI: 17.0–NA, #death events = 20). Finally, the mOS for 49 patients in the CD3D^high^/TGFB2^high^ mRNA group was 21.9 months (95% CI: 19.5–NA, #death events = 24). At low levels of CD3D mRNA expression, the increasing TGFB2 mRNA resulted in worse mOS outcomes; there was a significant difference in mOS outcomes between the CD3D^low^/TGFB2^low^ mRNA (*n* = 48, mOS = 66.9 months) and CD3D^low^/TGFB2^high^ mRNA (*n* = 40, mOS = 15.1 months, OS difference = −51.8, *p* = 0.0064) groups of patients (Figure 5A).

The mOS for 46 LCK^low^/TGFB2^low^ mRNA group patients was 66.9 months (95% CI: 21.4–NA, #death events = 17). The mOS for 42 patients in the LCK^low^/TGFB2^high^ mRNA group was 15.1 months (95% CI: 9.1–20.0, #death events = 29). Additionally, the mOS for 42 patients from the LCK^high^/TGFB2^low^ mRNA group was 17.0 months (95% CI: 15.5–49.4, #death events = 24). Lastly, the mOS for 47 patients in the LCK^high^/TGFB2^high^ mRNA group was 22.7 months (95% CI: 17.02–NA, #death events = 22). At low levels of LCK mRNA expression, increasing TGFB2 mRNA resulted in worse mOS outcomes; there was a significant difference in mOS outcomes comparing the LCK^low^/TGFB2^low^ mRNA (*n* = 46, mOS = 66.9 months) versus the LCK^low^/TGFB2^high^ mRNA (*n* = 42, mOS = 15.1 months, OS difference = −51.8, *p* = 0.0013) groups of patients. At low levels of TGFB2 mRNA, increasing LCK mRNA levels resulted in a worse prognosis; there was a significant difference in OS outcomes comparing the LCK^low^/TGFB2^low^ mRNA (*n* = 46, mOS = 66.9 months) versus the LCK^high^/TGFB2^low^ mRNA (*n* = 42, mOS = 17.0 months, OS difference = −49.9, *p* = 0.048) groups of patients. In contrast, at high levels of TGFB2 mRNA, increasing LCK mRNA levels improved the prognosis; there was a significant difference in OS outcomes comparing the LCK^low^/TGFB2^high^ mRNA (*n* = 42, mOS = 15.1 months) versus the LCK^high^/TGFB2^high^ mRNA (*n* = 47, mOS = 22.7 months, OS difference = 7.7, *p* = 0.027) groups of patients (Figure 5B).

The mOS for 47 HLA-DRA^low^/TGFB2^low^ mRNA group patients was 66.9 months (95% CI: 21.42–NA, #death events = 20). The mOS for 41 HLA-DRA^low^/TGFB2^high^ mRNA group patients was 15.1 months (95% CI: 10.1–22.5, #death events = 25). In the HLA-DRA^high^/TGFB2^low^ mRNA group, the mOS for 41 patients was 19.6 months (95% CI: 17.0–NA, #death events = 21). Lastly, the mOS for 48 patients in the HLA-DRA^high^/TGFB2^high^ mRNA group was 21.88 months (95% CI: 16.0–NA, #death events = 26). At low levels of HLA-DRA mRNA expression, increasing TGFB2 mRNA resulted in worse OS outcomes; there was a significant difference in OS outcomes comparing the HLA-DRA^low^/TGFB2^low^ mRNA (*n* = 47, mOS = 66.9 months) versus the HLA-DRA^low^/TGFB2^high^ mRNA (*n* = 41, mOS = 15.1 months, OS difference = −51.8, *p* = 0.020) groups of patients (Figure 5C).

The mOS for 44 patients in the RAC2^low^/TGFB2^low^ mRNA group was 66.9 months (95% CI: 20.2–NA, #death events = 17). The mOS for 44 patients in the RAC2^low^/TGFB2^high^ mRNA group was 17.9 months (95% CI: 12.4–22.7, #death events = 29), while those in the RAC2^high^/TGFB2^low^ (*n* = 44) mRNA group had an mOS of 18.7 months (95% CI: 16.4–49.4, death events = 24). Lastly, the RAC2^high^/TGFB2^high^ mRNA (n = 45) group recorded an mOS of 21.9 months (95% CI: 15.3–NA, #death events = 22). At low levels of RAC2 mRNA expression, increasing TGFB2 mRNA resulted in worse OS outcomes; there was a significant difference in OS outcomes between the RAC2^low^/TGFB2^low^ mRNA (*n* = 44, mOS = 66.9 months) and RAC2^low^/TGFB2^high^ mRNA (*n* = 44, mOS = 17.9 months, OS difference = −49, *p* = 0.018) groups of patients (Figure 5D).

The marked improvement in OS outcomes at low levels of TGFB2 and low levels of marker gene mRNA expression compared to high levels of TGFB2 mRNA expression in the TCGA dataset were confirmed for three of these genes (CD3D, LCK and RAC2) evaluable from the independent Kaplan–Meier plotter dataset (Figure 6). For the three marker genes, low levels of marker gene mRNA expression and low levels of TGFB2 mRNA both exhibited improved OS outcomes compared to high levels of TGFB2 mRNA (Figure 6A,C,E; HR ranged from 2.41 to 3.2 when comparing high versus low levels of TGFB2; the mOS ranged from 12.9 to 13.2 months for patients expressingTGFB2^high^ mRNA, improving to mOS times ranging between 21.5 to 35.9 months) confirming the results obtained from the TCGA dataset. There was no impact of TGFB2 mRNA at high levels of marker gene expression (Figure 6B,D,F).

We examined the prognostic OS impacts of *TGFB1* gene methylation in combination with the mRNA expression of the four marker genes (Appendix A). High levels of *TGFB1* methylation resulted in a trend of worse OS in combination with low-level expression of the marker genes. At low levels of LCK mRNA expression, increasing *TGFB1* gene methylation resulted in worse OS outcomes but did not achieve statistical significance. OS outcomes for patient groupings were as follows: LCK^low^/*TGFB1^lowMe^* (*n* = 40, mOS = 66.9 months); and LCK^low^/*TGFB1^highMe^* (*n* = 48, mOS = 17.9 months, OS difference = −49.0, *p* = 0.060) (Appendix A). There was a borderline significant difference in OS outcomes when comparing RAC2^low^/*TGFB1^lowMe^* (*n* = 40, mOS = 66.9 months) versus RAC2^low^/*TGFB1^highMe^* (*n* = 48, mOS = 19.4 months, OS difference = −47.4, *p* = 0.051) groups of patients (Appendix A). No significant differences were found in the survival curves considering the prognostic impact of *TGFB3* gene methylation in combination with marker gene expression levels in PDAC patients (Appendix A). When we examined combinations of *IFI27* gene methylation and the mRNA of marker genes (Appendix A), we found a significant positive prognostic impact of high levels of *IFI27* gene methylation at high levels of HLA-DRA mRNA expression. There was a significant difference in OS outcomes comparing the HLA-DRA^high^/*IFI27^lowMe^* (*n* = 34, mOS = 17.5 months) versus the HLA-DRA^high^/*IFI27^highMe^* (*n* = 55, mOS = 22.7 months, OS difference = 5.2, *p* = 0.039) groups of patients (Appendix A).

Our results also showed significant correlations between the expression of each biomarker and CD8^+^ T cell infiltration. The correlation coefficient values (Spearman rho) were as follows: 0.86 for CD3D; 0.64 for HLA DRA; 0.51 for LCK; and 0.38 for RAC2 (all comparisons *p* < 0.0001) (Appendix A).

We further investigated the effects of SMAD4 (Appendix A) to assess the impact of SMAD dependency [40], and SREBF1 to assess the impact of Gemcitabine resistance [41] (Appendix A) mRNA expression on the prognostic impact of *TGFB2* gene methylation. The favorable prognostic impact of *TGFB2* methylation (HR (95% CI range) = 0.513 (0.319–0.825); *p* = 0.006) was not affected by the inclusion of SMAD4 mRNA (HR (95% CI range) = 0.741 (0.518–1.061); *p* = 0.102) in the multivariate model. This model controlled for *TGFB1* methylation (HR (95% CI range) = 1.653 (1.081–2.528); *p* = 0.02); *TGFB3* methylation (HR (95% CI range) = 0.623 (0.405–0.958); *p* = 0.031), *IFI27* methylation (HR (95% CI range) = 0.525 (0.327–0.843); *p* = 0.008); age at diagnosis (HR (95% CI range) = 1.028 (1.007–1.05); *p* = 0.01); and sex (HR (95% CI range) = 0.838 (0.552–1.273); *p* = 0.407) (Appendix A). The inclusion of SREBF1 mRNA (HR (95% CI range) = 0.639 (0.44–0.926); *p* = 0.018) in the multivariate model resulted in a modest reduction in the positive prognostic impact of *TGFB2* gene methylation (HR (95% CI range) = 0.629 (0.39–1.016); *p* = 0.058) compared to the impact of *TGFB2* methylation evaluated in the model without SREBF1 mRNA levels (Figure 2: HR (95% CI range) = 0.53 (0.334–0.843); *p* = 0.007). The multivariate model that included SREBF1 mRNA controlled for the effects of *TGFB1* methylation (HR (95% CI range) = 1.43 (0.92–2.223); *p* = 0.112); *TGFB3* methylation (HR (95% CI range) = 0.629 (0.41–0.963); *p* = 0.033), *IFI27* methylation (HR (95% CI range) = 0.459 (0.291–0.723); *p* = 0.001); age at diagnosis (HR (95% CI range) = 1.029 (1.008–1.051); *p* = 0.007); and sex (HR (95% CI range) = 0.856 (0.566–1.294); *p* = 0.461) (Appendix A).

## 3. Discussion

### 3.1. Reactome Pathways Correlated to TGFB2 Gene Methylation

High levels of *TGFB2* gene methylation resulted in a significant improvement in OS in PDAC patients (HR (95% CI range) = 0.53 (0.334–0.843); *p* = 0.007), which was independent of *TGFB1/3*, *IFI27* methylations, age at diagnosis, and sex, as evaluated by the multivariate Cox proportional hazards model (Figure 2).

The multivariate Cox proportional hazards model that included the impact of SMAD4 to implicate the SMAD-dependent impact of TGFB ligands [40], showed no effect on the positive prognostic impact of *TGFB2* gene methylation (Appendix A), suggesting that the TGFB2 methylation was correlated to SMAD-independent pathways. Investigation of the Reactome pathways and mRNA expression of genes negatively correlated with *TGFB2* methylation identified three Reactome pathways (“Downstream TCR Signaling”, “CD28 Co-stimulation”, and the “CD28-Dependent Vav1 Pathway”) representing signaling cascades activated by T-cell engagement and were negatively correlated with TGFB2 gene methylation (*p* <0.01). This cascade is characterized by genes exhibiting greater than 20-fold increases in tumor tissues: HLA-DQB2 (Fold change = 97.3, *p* < 0.0001); HLA-DQA2 (Fold change = 77.6, *p* < 0.0001); HLA-DQA1 (Fold change = 59.2, *p* < 0.0001); TRAT1 (Fold change = 35.6, *p* < 0.0001); HLA-DRA (Fold change = 31, *p* < 0.0001); CD28 (Fold change = 29.7, *p* < 0.0001); HLA-DRB5 (Fold change = 29.7, *p* < 0.0001); CD3D (Fold change = 24.7, *p* < 0.0001); HLA-DQB1 (Fold change = 24.4, *p* < 0.0001); LCK (Fold change = 24.2, *p* < 0.0001); HLA-DRB1 (Fold change = 21.1, *p* < 0.0001); and HLA-DPA1 (Fold change = 20.8, *p* < 0.0001); PSMB9 (Fold change = 20.3, *p* < 0.0001).

Our findings suggest the identification of numerous potential diagnostic markers for susceptibility to T-cell receptor (TCR)-based therapies based on upregulation in tumor tissues. Recent studies have explored strategies to enhance T-cell receptor (TCR) signaling, including immune checkpoint inhibitors and adoptive T-cell therapies. These approaches aim to overcome immune evasion mechanisms and improve T-cell-mediated responses. For example, profiling the TCR repertoire has emerged as a potent method to investigate antitumor immune responses [42]. Additionally, immune checkpoint inhibition has shown promise in addressing the immunological barriers in the tumor microenvironment [43].

*TGFB2* gene methylation correlations were also represented by the expression of mRNA for genes associated with mesenchymal–epithelial transition pathways (MET) (“MET promotes cell motility”, “Syndecan interactions” and “Signaling by MET”) exhibited greater than 20-fold increases in expression in tumor tisues: COL11A1 (Fold change = 159.2, *p* < 0.0001); COL1A1 (Fold change = 83.5, *p* < 0.0001); COL3A1 (Fold change = 70.5, *p* < 0.0001); COL1A2 (Fold change = 56.1, *p* < 0.0001); FN1 (Fold change = 52.5, *p* < 0.0001); COL5A2 (Fold change = 40.6, *p* < 0.0001); COL5A1 (Fold change = 40.5, *p* < 0.0001); and TNC (Fold change = 28.7, *p* < 0.0001).

Recent studies have explored innovative approaches targeting the MET pathway, including the use of chimeric antigen receptor macrophages (CAR-M-c-MET) to inhibit pancreatic cancer progression [44]. Additionally, the MEK inhibitor Trametinib has shown promise in modifying the tumor microenvironment and enhancing the effectiveness of combined treatment regimens [45]. Our pathway analysis identified five genes with a greater than 50-fold increase in tumor tissues, facilitating the further development of diagnostic markers to identify patients susceptible to MET-targeting therapies.

### 3.2. Reactome Pathways Correlated to IFI27 Gene Methylation

There was a significant decrease in the hazard ratio (HR) for patients with high levels of *IFI27* gene methylation (HR (95% CI range) = 0.473 (0.302–0.74); *p* = 0.001), independent of *TGFB2* gene methylation (Figure 2). Evaluation of the Reactome pathways and the mRNA expression of genes correlated with *IFI27* gene methylation strongly represented Reactome pathways representing anaphase-promoting complex/cyclosome (APC/C), a ubiquitin–protein ligase involved in regulating the cell cycle (“Activation of APC/C and APC/C:Cdc20 mediated degradation of mitotic proteins”, “APC:Cdc20 mediated degradation of cell cycle proteins prior to satisfaction of the cell cycle checkpoint”, “APC/C:Cdc20 mediated degradation of Securin”, “APC/C-mediated degradation of cell cycle proteins”, “APC/C:Cdh1 mediated degradation of Cdc20 and other APC/C:Cdh1 targeted proteins in late mitosis/early G1”, “Regulation of APC/C activators between G1/S and early anaphase”, “Cdc20:Phospho-APC/C mediated degradation of Cyclin A”, and “Regulation of mitotic cell cycle”), were highly expressed and negatively correlated with *IFI27* gene methylation (*p* <0.01): NEK2 (Fold change = 170.4, *p* < 0.0001); UBE2C (Fold change = 150, *p* < 0.0001); CDC20 (Fold change = 62.9, *p* < 0.0001); PTTG1 (Fold change = 40.6, *p* < 0.0001); CDK1 (Fold change = 35.7, *p* < 0.0001); AURKB (Fold change = 34.2, *p* < 0.0001).

The spindle assembly checkpoint (SAC) tightly regulates the pathway, ensuring that chromosomes are correctly attached to the spindle before allowing for progression. Dysregulation of APC/C: Cdc20 can lead to genomic instability and is implicated in various diseases, including cancer [46,47]. Our analysis suggests that when considering both *TGFB2* and *IFI27* gene methylation as prognostic markers, the mRNA correlation with gene methylation yielded potential diagnostic markers markedly upregulated in tumor tissues.

### 3.3. Potential T-Cell-Associated Biomarkers Amplify the Impact of TGFB2 Gene Methylation

Patients with high levels of *TGFB2* methylation, when compared to low levels of *TGFB2* methylation, showed significant mOS improvements at low mRNA expression levels: 54.2 months for CD3 Delta Subunit of T-Cell Receptor Complex (CD3D) (*p* < 0.0001); 54 months for Lymphocyte-Specific Protein Tyrosine Kinase (LCK) (*p* = 0.0009); 54.9 months for Major Histocompatibility Complex, Class II, DR Alpha (HLA-DRA) (*p* = 0.0001); and 9 months for Rac Family Small GTPase 2 (RAC2) mRNA expression (*p* = 0.0057) (Figure 4). At low levels of *TGFB2* methylation, mOS was less than 13 months for CD3D, HLA DRA, and LCK, and 15.1 months for RAC2. At high levels of *TGFB2* methylation, mOS was almost 67 months for CD3D, HLA DRA, and LCK, and just over 24 months for RAC2 (Figure 4). *TGFB2* gene methylation drives TGFB2 mRNA to impact mOS, resulting in a mirrored response of methylation and mRNA. At low levels of TGFB2 mRNA expression, mOS was almost 67 months for all biomarkers, including RAC2; however, at high levels of TGFB2 mRNA expression, the mOS was between 15 and 18 months for all biomarkers (Figure 5). In contrast, the impact of high versus low *TGFB2* methylation on OS at high biomarker mRNA expression was insignificant. For all four markers, the difference between high and low levels of *TGFB2* methylation and high and low levels of TGFB2 mRNA expression was less than 6 months, with *p*-values greater than 0.05 (Figure 4 and Figure 5). Our results also showed significant correlations between the expressions of each of the biomarkers and CD8^+^ T cell infiltration. The correlation coefficient values (Spearman rho) were: 0.86 for CD3D, 0.64 for HLA DRA, 0.51 for LCK, and 0.38 for RAC2 (*p* < 0.0001) (Appendix A), suggesting that the most pronounced impact of high levels of *TGFB2* methylation and low levels of TGFB2 mRNA for improving prognostic outcomes occurs in immunosuppressed tumors with low levels of CD8^+^ T-cell infiltration.

Cytotoxic CD8^+^ T cells (CTLs) recognize and eliminate tumor cells through antigen recognition, activation, and targeted killing mechanisms. Tumor cells present antigens on major histocompatibility complex class I (MHC-I) molecules. These antigens include neoantigens (mutated proteins unique to cancer cells) and tumor-associated antigens (overexpressed or aberrantly expressed self-proteins) [48,49]. The T-cell receptor (TCR) on CD8^+^ T cells binds to these peptide-MHC-I complexes, initiating activation [50]. The CD8 co-receptor stabilizes TCR-MHC-I interaction to enhance downstream signaling [49]. Adhesion molecules (e.g., LFA-1, CD103) can strengthen cell–cell contact by binding ligands, such as ICAM-1 and E-cadherin, on tumor cells [50]. Full activation requires secondary signals (e.g., CD28-B7 interaction) from antigen-presenting cells such as dendritic cells [48]. CTLs also release perforin, which forms pores in the tumor cell membrane. Granzyme B can enter the tumor cell through these pores, activating caspase-dependent apoptosis [49,50]. In addition, the Fas ligand (FasL) on CTLs binds Fas receptors on tumor cells, triggering caspase-mediated apoptosis [49]. CTLs also release IFN-γ and TNF-α, which inhibit tumor proliferation [51]. Despite this, tumors can evade CD8^+^ T-cell immunity by downregulating MHC-I or antigen presentation and by expressing immune checkpoints (e.g., PD-L1) to induce T-cell exhaustion [49]. The therapies that aim to overcome these barriers include checkpoint inhibitors (e.g., anti-PD-1) and reinvigorating exhausted CTLs [49,50] and CAR-T cells engineered to target tumor antigens independently of MHC-I [49,51]. In the tumor microenvironment, high CD8^+^ T-cell infiltration correlates with better prognosis [49]; in ‘hot’ tumors, CTL-rich tumors respond better to immunotherapy [49,51] Our results suggest that TGFB2 mRNA blockade can improve the prognosis of patients with TMEs with low CD8^+^ T-cell infiltration, providing clinicians with an additional therapeutic modality.

The four suggested biomarkers from our investigation, which positively correlate with CD8^+^ T-cell infiltration and negatively correlate with *TGFB2* methylation in PDAC, have shown associations with prognosis across cancers including: breast cancer [52], bladder cancer [53] and head and neck squamous cell carcinoma [54].

CD3D expression in residual breast cancer tissue, after neoadjuvant chemotherapy, was associated with improved patient outcomes [52]. Furthermore, the CD3D/CD4 ratio was a prognostic marker in muscle-invasive bladder cancer, independently predicting better overall survival and recurrence-free survival [53].

HLA-DRA has been studied in cervical cancer [55,56], low-grade gliomas [57], renal clear cell carcinoma [58], pediatric adrenocortical tumors [59], non-small cell lung cancer [60], and breast cancer [61]. HLA-DRA expression was associated with increased disease-free and disease-specific survival, suggesting that HLA-DRA could be a promising biomarker for developing immunotherapies for cervical cancer [55]. Samuels et al. compared cervical squamous cell and cervical adenocarcinomas and found that HLA-DRA was expressed in 68.3% of squamous cell tumors and 93.8% of adenocarcinoma tumors and was associated with increased disease-free survival [56]. In their research into pediatric adrenocortical tumors (ACT), Leite et al. suggested that a lower expression of HLA-DRA, might contribute to more aggressive disease progression [59]. Pérez-Pena et al. reported that the increase in the expression of HLA-DRA in breast cancer, together with several other genes, was associated with high presence of CD8^+^ T Cells and improved patient outcomes [61]. Mei et al. researched non-small cell lung cancer and concluded that HLA-DRA could be a promising biomarker for NSCLC and a pan-cancer classifier for most immuno-hot tumors [60]. In contrast, in low-grade gliomas, HLA-DRA expression level was linked to immune infiltration; a high expression of HLA-DRA was associated with a poor prognosis [57].

LCK has been implicated in leukemia, breast cancer, lung cancer, bile duct cancer, gliomas, colorectal cancer, and melanomas [62]. LCK was reported to be important in the functionality of T-cells and the role of LCK in the series of signaling events that occur within a T-cell following the binding of its T-cell receptor (TCR) to a specific antigen [62,63,64]. Due to LCK’s essential function in T-cell responses, various strategies have been devised to redirect LCK activity, to enhance the efficacy of chimeric antigen receptors (CARs) and augment T-cell responses in the context of cancer immunotherapy [65].

RAC2 is part of an immune-related signature for breast cancer. RAC2 was one of the nine genes (C14orf79, C1orf168, C1orf226, CELSR2, FABP7, FGFBP1, KLRB1, PLEKHO1, and RAC2) identified as part of the immune-related signature. One study found that higher expression of RAC2, along with the other eight genes in the signature, was correlated with a better prognosis in breast cancer patients [66]. Chen Q et al. reported that RAC2, identified alongside APOBEC3D and TNFRSF14 in a proptosis-related three-gene signature, is a prognostic marker in breast cancer. Lower RAC2 expression correlates with poorer outcomes, as higher expression is protective [67]. Xu Y et al. reported that RAC2 is associated with T cell infiltration and tumor suppression in breast cancer and is potentially a biomarker [68].

We also considered the possible effect of specific TGFB2 mRNA knockdown (high levels of *TGFB2* methylation or low levels of TGFB2 mRNA) compared to the impact of the combined effect of the four prognostic markers and *TGFB1/3* or *IFI27* gene methylations. Our results (Appendix A) suggest that *TGFB3* methylation does not significantly affect overall survival on either low or high expression of each of the suggested biomarkers. Conversely, *TGFB1* methylation levels do have an effect. However, knockdown of TGFB1 via high levels of *TGFB1* methylation showed a worse prognosis, whereas the opposite effect was observed for knockdown of TGFB2 via high levels of *TGFB2* gene methylation. At low levels of LCK mRNA expression, for example, the difference in mOS between low (mOS = 66.9 months) and high (mOS = 17.9 months) *TGFB1* gene methylation was borderline significant at 49 months (*p* = 0.06). Similar results were obtained for the *TGFB1* methylation and RAC2 mRNA combination. There was a borderline significant difference in OS outcomes for comparing the RAC2^low^/*TGFB1^lowMe^* (*n* = 40, mOS = 66.89 months) versus the RAC2^low^/*TGFB1^highMe^* (*n* = 48, mOS = 19.45 months, OS difference = −47.44, *p* = 0.051) groups of patients.

These results imply that targeting the TGFB2 isoform, specifically when CD8^+^ T-cell infiltration is low, could provide therapeutic benefits for PDAC patients. Our studies indicate that OT-101 (Trabedersen), a targeted antisense molecule, binds to human TGFB2 mRNA. A Phase I/II trial demonstrated that patients with PDAC treated with OT-101 followed by chemotherapy experienced improved overall survival (OS). OT-101 works by inhibiting TGFB signaling [69]. Recently, TGFB2 has been shown to confer Gemcitabine resistance by upregulating the lipogenesis regulator sterol regulatory element binding factor 1 (SREBF1) [41]. The inclusion of SREBF1 mRNA (HR (95% CI range) = 0.639 (0.44–0.926); *p* = 0.018) in the multivariate model resulted in a modest reduction in the positive prognostic impact of *TGFB2* gene methylation (Appendix A: HR = 0.629; *p* = 0.058) compared to the impact of *TGFB2* methylation evaluated in the model without SREBF1 mRNA levels (Figure 2: HR = 0.53; *p* = 0.007) [41]. These findings suggest that OT-101 may enhance clinical outcomes in PDAC, especially in tumors with low levels of T-cell infiltration. A clinical study is underway to compare the efficacy and safety of OT-101 combined with FOLFIRINOX to FOLFIRINOX alone in patients with advanced or metastatic pancreatic cancer with no planned treatment with Gemcitabine (NCT06079346).

The present study has significant limitations, as the bioinformatics-based analyses were used without additional supportive laboratory testing of *TGFB1/2/3* and *IFI27* gene methylation and mRNA levels. Validating mRNA levels of T-cell markers will require platforms, such as quantitative RT-PCR and immunohistochemistry, to identify CD8^+^ T-cell populations. Our findings, which show a correlation between gene methylation levels and survival outcomes, will require further experimental validation using PDAC tumor biopsies in future clinical trials that monitor the biochemical components identified in these studies. Monitoring DNA methylation offers a more robust biomarker due to its chemical stability and a direct link to gene regulation, as evidenced by mRNA monitoring, which enables the characterization of the tumor microenvironment and the identification of patients for immunotherapies in cold tumors.

## 4. Materials and Methods

### 4.1. Domain-Specific Identification of PubMed Articles Augmented by Artificial Intelligence

PubMed searches using the keywords “Pancreatic AND prognosis AND TCGA” retrieved 1195 abstracts; “Pancreatic AND adenocarcinoma AND methylation” (995 abstracts); and “Pancreatic AND single-cell RNA-seq” (1137 abstracts) (https://pubmed.ncbi.nlm.nih.gov/ accessed 16 December 2024) were downloaded as text documents for processing using the Chatbot-enabled tools developed at Oncotelic Therapeutics. Each abstract was then (aided using puppeteer 19.11.1) embedded and transformed (langchain-openai 0.2.3, openai 1.52.0) into a vector of numbers capturing semantic similarity between text elements (tokens) and then stored in our Qdrant vector database (https://qdrant.tech/ accessed 16 December 2024). Semantically similar abstracts were transformed into the same vector “embedding” space, as the embedding has been trained to minimize the distance between pairs of abstracts in this space. Using an agglomerative clustering algorithm (hdbscan 0.8.39) to group the vectors, we automatically labeled these clusters using the question-answering model to identify any similarity between the abstracts corresponding to each cluster’s vectors. During the question-answering process, the user’s query is converted into an embedding vector. A similarity metric, such as cosine similarity, is then utilized to find the embedded abstract vectors nearest to the vector representing the query. The abstracts that correspond to these nearest vectors are subsequently provided to the question-answering model as context, alongside the original query, to produce a response to the user’s question.

React framework served as the backbone of the user interface, providing an open-source and flexible solution for developing powerful front-end user-interfaces. (https://react.dev/ accessed 25 March 2024). In addition, we used the @mui/material (https://mui.com/ accessed on 25 March 2024)) libraries for the interface’s design aspects and aimed to follow the material design guidelines closely. Serving the front end was Node.Js (https://nodejs.org/en accessed 25 March 2024). The Node.js libraries included in the project were @adobe/pdfservices-node-sdk 3.4.2@aws-sdk/client-s3 3.412.0, @langchain/community 0.2.5, @material-ui/core 4.12.4, @mui/base 5.0.0-beta.18, @mui/icons-material 5.11.16, @mui/material 5.15.20, @mui/styled-engine-sc 5.12.0, @mui/x-date-pickers 6.15.0, @qdrant/js-client-rest 1.4.0, carrot2 0.0.1, pdf2img 0.5.0, pdfjs-dist 4.5.136, puppeteer 19.11.1, react 18.2.0, sequelize 6.31.1, and zod 3.22.4.

The integration of our Chatbot technology applied to PubMed abstracts facilitated the rapid discovery of key primary publications used in the writing of this manuscript.

### 4.2. Demethylation Impact of TGFB1/2/3 and IFI27 Genes on Overall Survival (OS) for Pancreatic Adenocarcinoma Patients (PDAC)

We accessed methylation (HM450) beta values for genes in 195 cases (Web portal: https://www.cbioportal.org/study/summary?id=paad_tcga, accessed 1 February 2024 for values reported in the “data_methylation_hm450.txt” data file accessed 1 February 2024) of which clinical overall survival (OS) outcomes were reported for 178 patients. The patients were stratified according to high (*n* = 89) and low (*n* = 89) gene methylation levels using the median cut-off values (≥median cut-off for) for each of the four genes investigated (*TGFB1/2/3* and *IFI27*). Kaplan–Meier analysis generated OS curves for the stratified patient groupings to investigate the prognostic impact of patients with high versus low *TGFB1/2/3* and *IFI27* methylation levels. Statistical significance between stratified patient groupings was tested using the log-rank chi-square test, implemented utilizing R-based software packages including survival_3.2-13, survminer_0.4.9, and survMisc_0.5.5. To present the treatment outcomes in a graphical format, we plotted graphs using dplyr_1.0.7, ggplot2_3.3.5, and ggthemes_4.2.4 implemented in R.

### 4.3. Hazard Ratio (HR) Comparisons for PDAC Patients to Determine the Independent Effects of High Levels of TGFB1/2/3, IFI27 Gene Methylation Levels, Age, Sex

Multivariate analyses utilized the Cox proportional hazards model to assess the individual effects of *TGFB1/23* and *IFI27* gene methylations on mOS (*n* = 178, 93 death events). This analysis controlled for age at diagnosis and sex. Briefly, the model included: (i) methylated *TGFB2* levels (median cut-off); (ii) methylated *TGFB1* levels (median cut-off); (iii) methylated *TGFB3* levels (median cut-off); (iv) methylated *IFI27* levels (median cut-off); (v) age at diagnosis as a linear covariate; and (vi) sex. All calculations were performed using the R platform (survival_3.2–13 ran in R version 4.1.2). Hazard ratios were estimated using the exponentiated regression coefficients and were visualized using forest plots (survival_3.2–13 and survminer_0.4.9 ran in R version 4.1.2 (1 November 2021)).

### 4.4. Geneset Enrichment Analysis (GSEA) in Reactome Pathways to Identify Negatively Correlated Pathways to TGB1/2/3 and IFI27 Methylation Levels

Beta values for *TGFB1/2/3* and *IFI27* gene methylation were correlated using Spearman ranks with mRNA expression levels of 14,861 genes for all PDAC patients (*n* = 177 evaluable patients) across 1286 Reactome pathways. Normalized enrichment scores (NESs) for the Reactome pathways were computed from the ranked correlation coefficients (implemented using the fgsea version 1.20.0 package in R). The permutation *p*-values for normalized enrichment scores were determined, identifying pathways significantly correlated with *TGFB1/2/3* and *IFI27* methylation. We used a two-way hierarchical clustering technique to organize the NES scores for the most affected pathways (*p* < 0.0001) grouped according to the average distance metric (default Euclidean distance), identifying columns of gene methylations and rows of Reactome pathways (heatmap.2 function in the R package gplots_3.1.1). Three clusters of Reactome pathways were identified to compile a list of genes for further processing: i. Pathways negatively correlated with *TGFB2/3* and *IFI27* gene methylations and positively correlated with *TGFB1* gene methylation; ii. Pathways negatively correlated with IFI27 gene methylations; and iii. TGFB2-specific gene methylation (*p*-value TGFB2 methylation < 0.05 & NES TGFB2 methylation < 0 & *p*-value TGFB1 methylation > 0.1 & *p*-value TGFB3 methylation > 0.1 & *p*-value IFI27 methylation > 0.1). We then identified genes whose mRNA levels were negatively correlated with either TGFB2 or IFI27 gene methylation (*p* < 0.01, FDR < 0.1) for determining tumor-specific markers, comparing mRNA expression in normal versus tumor tissues.

### 4.5. Differential Expression of mRNA Comparing PDAC Tumors Versus Normal Pancreatic Tissue Samples

We utilized log_2_ transformed transcripts per million (TPM) summarized RNAseq data files (https://xenabrowser.net/datapages/?dataset=TcgaTargetGtex_rsem_gene_tpm&host=https%3A%2F%2Ftoil.xenahubs.net&removeHub=https%3A%2F%2Fxena.treehouse.gi.ucsc.edu%3A443, accessed on 25 July 2023 and then downloading onto the desk top following the link “https://toil-xena-hub.s3.us-east-1.amazonaws.com/download/TcgaTargetGtex_rsem_gene_tpm.gz; Full metadata”, accessed on 25 July 2023) downloaded from the UCSC Xena web platform (https://xenabrowser.net/datapages/ accessed on 25 July 2023) to compare gene expression levels for 178 tumor tissue samples (search term: “TCGA Pancreatic Adenocarcinoma”) versus 167 pancreatic tissue samples (search term: “GTEX Pancreas”). This resource reports results from the UCSC Toil RNAseq recompute compendium, which is a standardized, realigned and recalculated gene and transcript expression dataset for all TCGA, and GTEx, which enables users to contrast gene and transcript expression between TCGA “tumor” samples and corresponding GTEx “normal” samples [70]. We applied a two-way ANOVA model to identify differentially expressed genes to compare normal versus tumor tissue samples. The log_2_-transformed TPM values for Gene and Tissue were included as fixed factors, along with one interaction term to investigate gene-level effects for normal and tumor tissues (Gene × Tissue). For each gene, we conducted a comparison between normal and tumor samples blocked by the Gene factor and then determined significance by adjusting the *p*-value using the false discovery rate algorithm provided in the R-package (FDR-corrected for all pairs) calculations performed in R using multcomp_1.4–17 and emmeans_1.7.0 packages ran in R version 4.1.2 (1 November 2021) with RStudio front end (RStudio 2021.09.0 + 351 “Ghost Orchid” Release). Bar chart graphics were constructed using the ggplot2_3.3.5 R package.

We used a two-way hierarchical clustering technique to organize expression patterns. Samples and gene expressions displaying similar expression profiles were grouped together using the average distance metric (default Euclidean distance implemented using the heatmap.2 function in the R package gplots_3.1.1). The cluster figure displayed the mean expression levels in tumor tissue centered on normal pancreas expression levels, representing log_2_-transformed Fold change values. The associated dendrograms organized and depicted expression levels of co-regulated genes for both (rows) and for pancreatic cancer patients (columns).

### 4.6. Using Multivariate Cox Proportional Hazards Models, TGFB2 Gene Methylation Dependency on Marker Gene mRNA

Batch-normalized mRNA expression data for 177 patients diagnosed with PDAC were accessed via the cBioportal repository (https://www.cbioportal.org/study/summary?id=paad_tcga_pan_can_atlas_2018, accessed on 13 August 2022) for the mRNA expression of marker genes. Multivariate analyses utilized the Cox proportional hazards model to assess the individual effects of *TGFB1/23* and *IFI27* gene methylations on OS (*n* = 177, 92 death events) and the impact of marker genes (greater than 20-fold increase in mRNA expression in tumor tissues and negatively correlated with *TGFB2* or *IFI27* gene methylations) and for determining the *TGFB2* gene methylation dependency using the statistical interaction term from the model of the marker gene by *TGFB2* gene methylation. This analysis controlled for age at diagnosis and sex. Briefly, the model included the HR calculations of: (i) marker gene expression (Z-scores of TPM levels); (ii) methylated *TGFB2* levels (median cut-off); (iii) methylated *TGFB1* levels (median cut-off); (iv) methylated *TGFB3* levels (median cut-off); (v) methylated *IFI27* levels (median cut-off); (vi) age at diagnosis as a linear covariate; (vii) sex; and (viii) *TGFB2* gene methylation × Marker gene mRNA expression interaction term.

We further evaluated the marker genes that showed a significant interaction term with *TGFB2* methylation using Kaplan–Meier analysis of four groups of patients stratified according to median expression of the marker gene in combination with the median cut-off of *TGFB1/2/3* and *IFI27* gene methylations. To test whether *TGFB2* methylation also drove the impact of TGFB2 mRNA, we also compared high versus low levels of TGFB2 mRNA paired with the four marker genes. Using the log-rank chi-square test, we tested for statistical significance, implemented utilizing R-based software packages including survival_3.2-13, survminer_0.4.9, and survMisc_0.5.5. To present the treatment outcomes in a graphical format, we plotted graphs using dplyr_1.0.7, ggplot2_3.3.5, and ggthemes_4.2.4 implemented in R. We considered *p*-values lower than 0.05 significant after adjusting for multiple comparisons across four groups (6 comparisons) using the Benjamini and Hochberg method.

### 4.7. Marker Gene mRNA Expression Correlation with T-Cell Infiltration into PDAC Tumors

We estimated the correlation of marker gene mRNA expression levels (RSEM estimated log_2_TPM values) and T-cell immune-cell infiltration in PDAC tumors using the algorithms provided in the TIMER2.0 (http://timer.cistrome.org/, accessed 30 January 2025) [71] web tool compiled for The Cancer Genome Atlas (TCGA). The purity-adjusted Spearman’s rho correlations utilizing the CIBERSORT-ABS [72] immune deconvolution method was used to estimate T-cell infiltration in PDAC tumors.

### 4.8. Confirmation of the Prognostic Impact of Marker Genes Identified from the TCGA Dataset

Three out of the four prognostic markers identified from the TCGA dataset (CD3D, LCK and RAC2) were available for evaluation in the Kaplan–Meier Plotter web portal made accessible via a commercial license (https://kmplot.com/private/index.php?cancer=pancreas_rma&p=service, accessed on 26 May 2025). The public version of the portal can be accessed via https://server2.kmplot.com/pancreas (accessed on 26 May 2025). This pancreatic cancer database was compiled from publicly available GEO data and further processed for survival analysis using the web portal [73,74].

## 5. Conclusions

This study analyzed the methylation of *TGFB2*, *TGFB3*, and *IFI27* genes using single-gene markers and median cut-off values for Kaplan–Meier plots. The following modest improvements in median overall survival (OS) were noted: 5.7 months for *TGFB2* (*p* = 0.044); 5.2 months for *IFI27* (*p* = 0.036); and 3.7 months for *TGFB3* (*p* = 0.028) methylation. Conversely, high *TGFB1* methylation led to a reduced median OS of 4.7 months (*p* = 0.016). Multivariate Cox proportional hazards analysis confirmed that the prognostic significance of *TGFB1*, *TGFB2*, *TGFB3*, and *IFI27* methylations was independent. High *TGFB2* methylation was notably associated with improved OS in pancreatic ductal adenocarcinoma (PDAC) patients, showing a hazard ratio (HR) of 0.53 (95% CI: 0.334–0.843; *p* = 0.007). Survival benefits from *TGFB2* methylation increased at the low expression of CD8^+^ T cell-associated markers. Significant OS improvements can be achieved at low expression levels for CD3D (54.2 months, *p* < 0.0001), LCK (54 months, *p* = 0.0009), HLA-DRA (54.9 months, *p* = 0.0001), and RAC2 (9 months, *p* = 0.0057), comparing high versus low levels of *TGFB2* methylation. Therefore, *TGFB2* methylation is a key prognostic marker for PDAC, particularly in immunosuppressed microenvironments with low CD8^+^ T-cell infiltration. This correlation suggests that targeting TGFB2 mRNA with knockdown strategies, such as OT-101, could enhance PDAC prognosis. Monitoring DNA methylation of *TGFB1*, *TGFB2*, and *TGFB3* provides a robust method for characterizing tumor microenvironments and selecting candidates for immunotherapy in cold tumors.

## Figures and Tables

**Figure 1 ijms-26-05567-f001:**
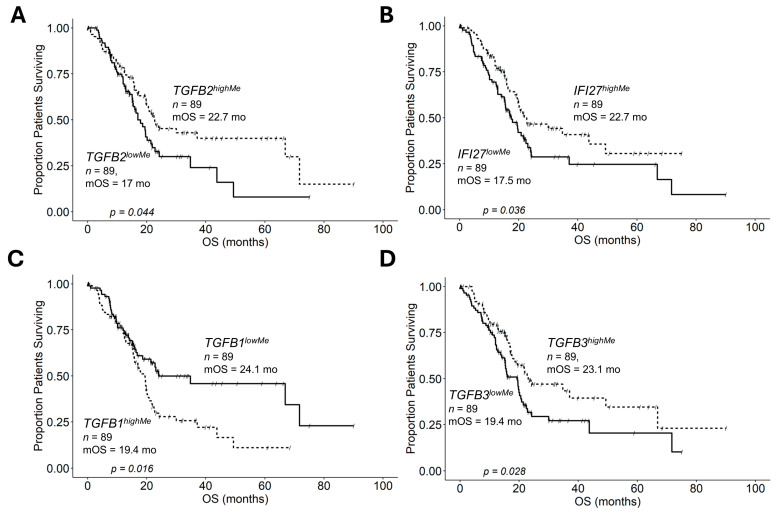
High levels of *TGFB2/3* and *IFI27* gene methylations, not *TGFB1* methylations, improve overall survival (OS) in PDAC patients. PDAC patients were stratified according to high (highMe) and low (lowMe) levels of gene methylation using the median cut-off values (≥median cut-off for highMe) for each of the four genes investigated (*TGFB1/2/3* and *IFI27*). Kaplan–Meier analysis generated OS curves for the stratified patient groupings to investigate the prognostic impact of patients with high versus low *TGFB1/2/3* and *IFI27* methylation levels: (**A**) The *TGFB2^highMe^* group of patients (mOS = 22.7 months) exhibited a significantly longer OS outcome than *TGFB2^lowMe^* group of patients (mOS = 17.0 months, *p* = 0.044); (**B**) the *IFI27^highMe^* group of patients (mOS = 22.7 months) exhibited a significantly longer OS outcome than *IFI27^lowMe^* patients (mOS = 17.5 months, *p* = 0.036); (**C**) the *TGFB1^highMe^* group of patients (mOS = 19.4 months) exhibited a significantly shorter mOS outcome than the *TGFB1^lowMe^* group of patients (mOS = 24.1, *p* = 0.016); and (**D**) the *TGFB3^highMe^* group of patients (mOS = 23.1 months) exhibited a significantly longer OS outcome than the *TGFB3^lowMe^* group of patients (mOS = 19.4, *p* = 0.028).

**Figure 2 ijms-26-05567-f002:**
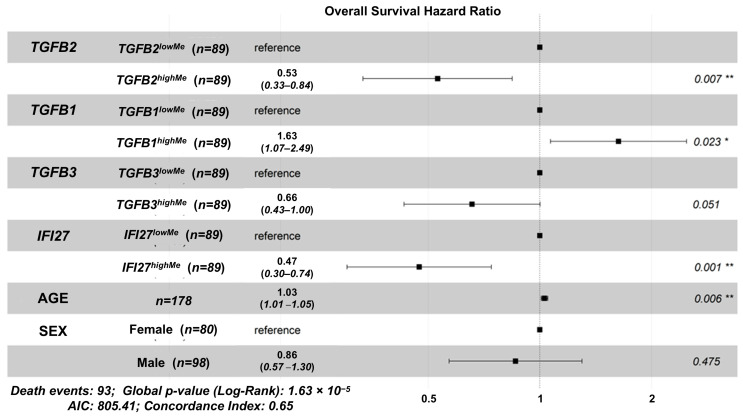
Independent impacts of gene methylations, controlling for age and sex, using the Cox proportional hazards model to assess OS. We used the multivariate model to further substantiate the impact of gene methylations on OS as independent variables, considering their correlation with each other and other variables, including age and sex. The forest plot depicts the hazard ratios (HR; black boxes) for each variable tested with 95% confidence intervals. * denotes *p* < 0.05, and ** denotes *p* < 0.01.

**Figure 3 ijms-26-05567-f003:**
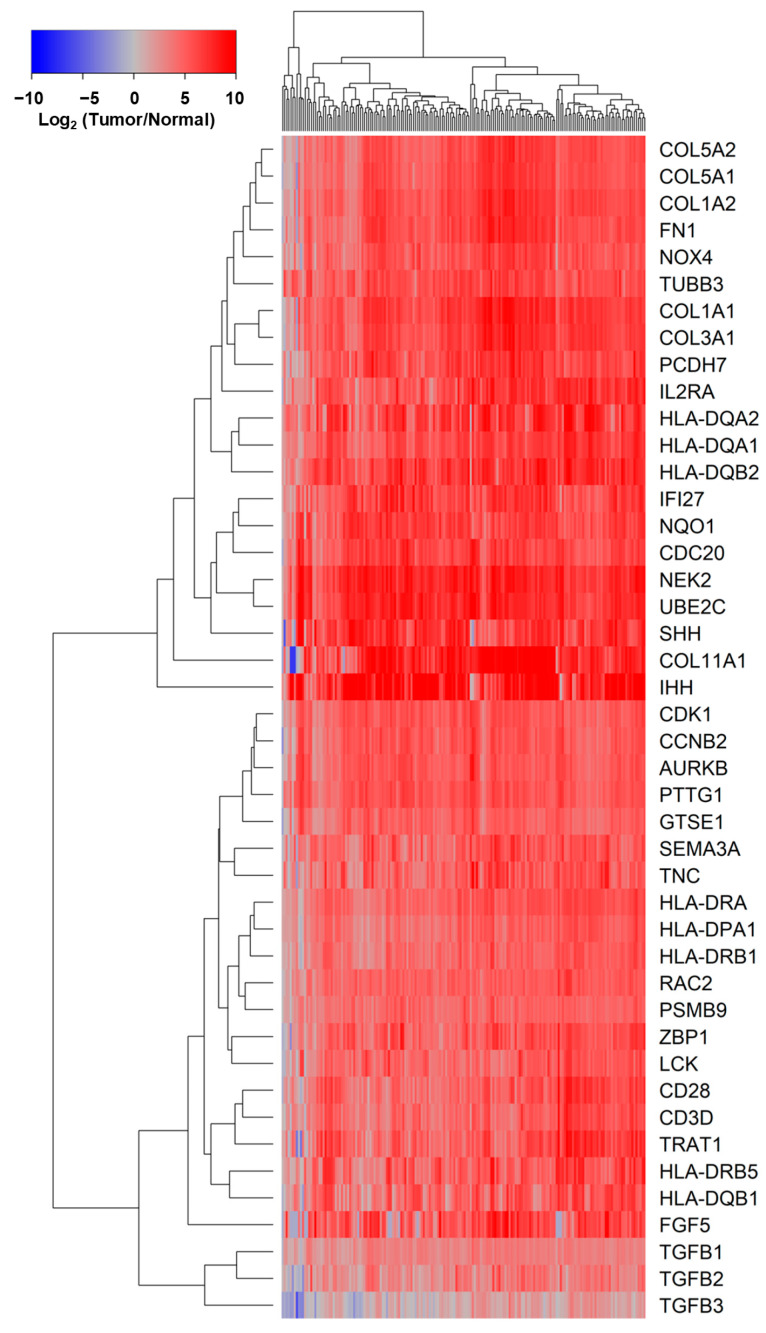
Upregulation of genes negatively correlated with *TGFB2* or *IFI27* gene methylation. We compiled 856 genes by combining the gene lists from Appendix A that identified 358 genes negatively correlated (*p* < 0.01, FDR = 0.06) to *TGFB2* or *IFI27* gene methylations to determine upregulated genes in tumor versus normal tissue comparisons. Of these 356 genes, 173 exhibited a greater than 5-fold increase in expression in tumor tissue (*p* < 0.0001, FDR < 0.0001); 87 genes exhibited a greater than 10-fold increase in expression (*p* < 0.0001, FDR < 0.0001); and 41 genes exhibited a greater than 20-fold increase in expression (Appendix A). The cluster figure depicts log_2_-transformed patient-level Fold change values for tumor (*n* = 178 evaluable mRNA levels), mean-centered to expression in normal tissue (*n* = 167) comparisons organized using a two-way hierarchical clustering algorithm (blue to red for increasing Fold change). The figure includes 41 genes upregulated in tumor tissue (>20-fold) and 3 genes for TGFB1/2/3 mRNA.

**Figure 4 ijms-26-05567-f004:**
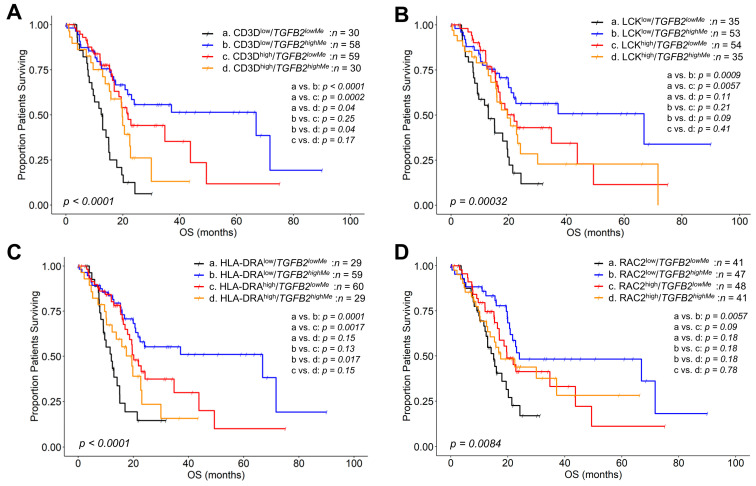
Favorable prognostic OS impacts of high levels of *TGFB2* gene methylation at low marker gene expression levels in PDAC patients. PDAC patients were correlated with OS outcomes by investigating the impact of methylation beta values (median cut-off for high methylation levels; superscripted “highMe” compared to low methylation; “lowMe” for *TGFB2* methylation). These patients were further stratified into four groups based on gene expression levels of: CD3D (**A**); LCK (**B**); HLA-DRA (**C**); and RAC2 (**D**) mRNA expression levels (median cut-off for low and high values). The Kaplan–Meier plots display four stratified curves for each of the marker genes. Six pairwise comparisons were performed between the four groups of patients (*p*-value adjusted using the BH correction). The four marker genes, low marker gene mRNA expression levels, and low levels of *TGFB2* methylation (black lines) all exhibited worse OS outcomes than low levels of marker gene and high levels of *TGFB2* methylation (blue lines).

**Figure 5 ijms-26-05567-f005:**
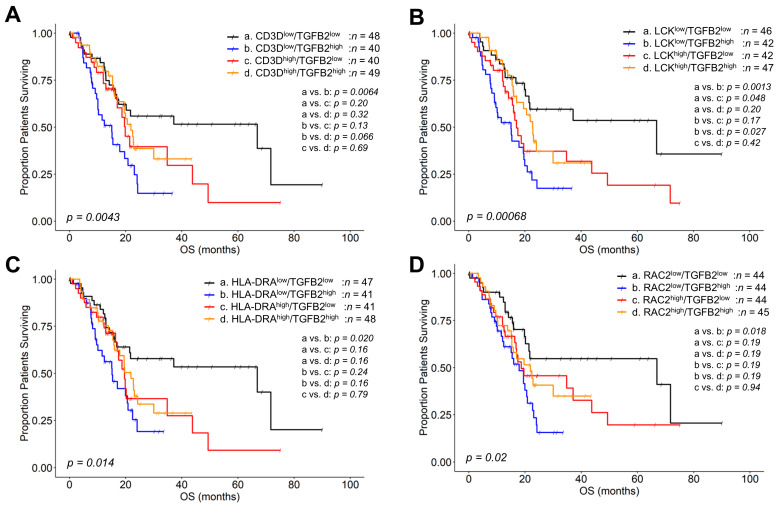
Negative prognostic OS impacts of high levels of TGFB2 mRNA expression at low marker gene expression levels in PDAC patients. PDAC patients were correlated to OS outcomes investigating the impact of mRNA expression of TGFB2 (median cut-off for high and low levels of expression), further stratified into four groups based on gene expression levels of CD*3D* (**A**); LCK (**B**); HLA-DRA (**C**); and RAC2 (**D**) mRNA expression levels (median cut-off for high and low levels) in these patients. The Kaplan–Meier plots show four stratified curves for each of the marker genes. For the four marker genes, low levels of marker gene mRNA expression and low levels of TGFB2 mRNA (black lines) all exhibited improved OS outcomes compared to low levels of marker gene and high levels of TGFB2 mRNA (blue lines). These results suggest that the *TGFB2* methylation drives TGFB2 mRNA expression to impact OS outcomes in combination with the four marker genes.

**Figure 6 ijms-26-05567-f006:**
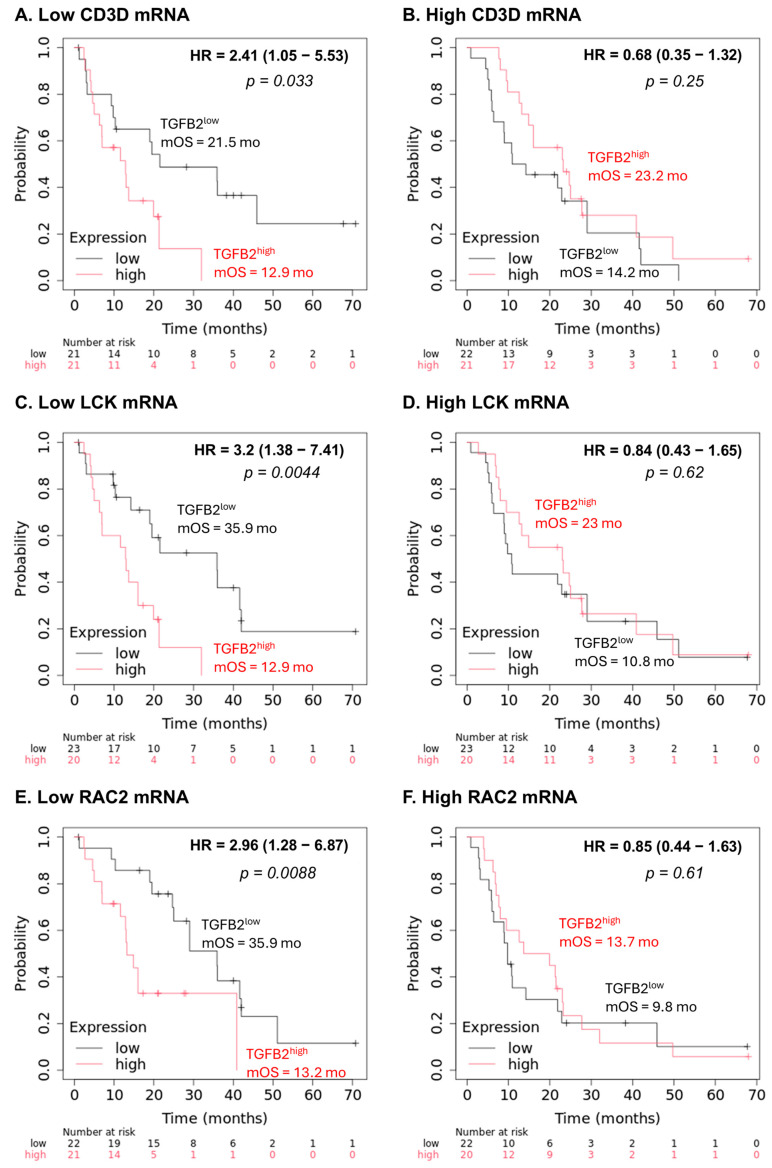
Confirmation using an independent dataset reveals that negative prognostic OS impacts are associated with high levels of TGFB2 mRNA expression at low marker gene expression levels in PDAC patients. Three out of the four prognostic markers identified from the TCGA dataset were available for evaluation; all three were confirmed for their prognostic impact using an independent dataset of 86 patients screened with the Affymetrix platform and deposited in the Kaplan–Meier plotter database. PDAC patients were correlated to OS outcomes investigating the impact of mRNA expression of TGFB2 (median cut-off for high and low levels of expression), further stratified based on gene expression levels of: CD*3D* (**A**,**B**); LCK (**C**,**D**); and RAC2 (**E**,**F**) mRNA expression levels.

## Data Availability

Methylation (HM450) beta values for genes are available at Web portal: https://www.cbioportal.org/study/summary?id=paad_tcga (accessed 1 February 2024 for values reported in the “data_methylation_hm450.txt” data file accessed 1 February 2024). We utilized log_2_ transformed transcripts per million (TPM) summarized RNAseq data files (https://xenabrowser.net/datapages/?dataset=TcgaTargetGtex_rsem_gene_tpm&host=https%3A%2F%2Ftoil.xenahubs.net&removeHub=https%3A%2F%2Fxena.treehouse.gi.ucsc.edu%3A443, accessed on 25 July 2023 and then downloading onto the desk top following the link https://toil-xena-hub.s3.us-east-1.amazonaws.com/download/TcgaTargetGtex_rsem_gene_tpm.gz; Full metadata, accessed on 25 July 2023) downloaded from the UCSC Xena web platform (https://xenabrowser.net/datapages/ accessed on 25 July 2023) to compare gene expression levels for 178 tumor tissue samples (search term: “TCGA Pancreatic Adenocarcinoma”) versus 167 pancreatic tissue samples (search term: “GTEX Pancreas”). Batch-normalized mRNA expression data for 177 patients diagnosed with PDAC for the mRNA expression of marker genes were accessed from the cBioportal repository (https://www.cbioportal.org/study/summary?id=paad_tcga_pan_can_atlas_2018, accessed on 13 August 2022). T-cell immune-cell infiltration values in PDAC tumors were accessed and calculated using the algorithms provided in the TIMER2.0 (http://timer.cistrome.org/ accessed 30 January 2025) web portal.

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
