# Peer review of "TGFB2* Gene Methylation in Tumors with Low CD8^+^ T-Cell Infiltration Drives Positive Prognostic Overall Survival Responses in Pancreatic Ductal Adenocarcinoma"

_ijms, 2025, doi:10.3390/ijms26125567_

Round 1

Reviewer 1 Report

Comments and Suggestions for Authors

This manuscript analyzed TGFB2 gene methylation in PDAC by bioinformatic analysis and found that TGFB2 gene methylation is associated with low CD8+ T-cell infiltration and drives positive prognostic overall survival responses. It is interesting, but should be improved.

  1. Figure 1, 4 and 5,how many patients in each group?They should be indicated.
  2. Methylation has significant effect on transcriptional levels of TGFB1/2/3 and IFI27 gene. Authors should analyze the correlation between methylation and mRNA levels of these genes in patients.
  3. The association between the mRNA expression of TGFB1/2/3 and IFI27 gene and patients’ overall survival should be analyzed.
  4. The correlation between TGFB2 mRNA and marker genes’ expression in patients should be analyzed.
  5. “5. Conclusions” is too long. Try to make conclusion in a paragraph briefly and precisely.
  6. The analyses only depend on TCGA PDAC data. Their own PDAC cohort or datasets in GEO should be included to confirm the findings from TCGA database.

Author Response

1.Figure 1, 4 and 5,how many patients in each group?They should be indicated.

We have now included patient numbers in all these figures.

2.Methylation has significant effect on transcriptional levels of TGFB1/2/3 and IFI27 gene. Authors should analyze the correlation between methylation and mRNA levels of these genes in patients.

We thank the reviewer for this suggestion to assess the correlations between methylation and mRNA expression of the genes of interest.  The updated manuscript now includes supplementary figure 1 with the following text in the main document:

“To assess whether gene methylation can functionally drive the mRNA product, resulting in a prognostic impact in PDAC patients, we first correlated gene methylation with the corresponding mRNA (Figure S1). Then we evaluated the prognostic impact of mRNA levels on overall survival (OS) (Figure S2). Greater than 30% of the variation was explained for the positive correlation of TGFB2 (R2 = 0.316, p < 0.001), IFI27 (R2 = 0.356, p < 0.001), and TGFB3 (R2 = 0.322, p < 0.001) methylation with the corresponding mRNA. TGFB1 gene methylation exhibited a weak correlation with TGFB1 mRNA (R2 = 0.05, p < 0.03). These results suggest that TGFB2, IFI27, and TGFB3 gene methylations can drive mRNA products for the corresponding genes (Figure S1). “ (Line 177-86)

3.The association between the mRNA expression of TGFB1/2/3 and IFI27 gene and patients’ overall survival should be analyzed.

The updated manuscript now includes supplementary figure 2 with the following text in the main document:

“The correlations between TGFB2 and IFI27 gene methylation and mRNA expression were prognostically associated with OS outcomes, comparing high versus low expression.  This was not observed for the correlation between TGFB1 and TGFB3 mRNA and median OS outcomes (mOS) (Figure S2).  TGFB2high subset of patients (mOS = 19.4; 95% CI = 15.34 - 22.71 months; n = 89; # death events = 51) exhibited a significantly shorter OS outcome than TGFB2low patients (mOS = 21.71; 95% CI = 18.66 - NA months; n = 88; #death events = 41, p = 0.034) (Figure S2A). IFI27high patients (mOS = 16.8; 95% CI = 15.12 - 22.47 months; N = 89; #death events = 52) exhibited a significantly shorter OS outcome than IFI27low patients (mOS = 24.05; 95% CI = 19.65 - NA months; n = 88; #Events = 40; p = 0.009) (Figure S2B).” (Lines  185-94)

4.The correlation between TGFB2 mRNA and marker genes’ expression in patients should be analyzed.

The updated manuscript now includes supplementary figure 6 with the following text in the main document:

“Weak correlations were observed for CD3D (< 10% of the explained variation; R2 = 0.042, p = 0.007), LCK mRNA (R2 = 0.021, p = 0.054), and RAC2 (R2 = 0.079, p < 0.001) with TGFB2 mRNA. Correlation of HLA-DRA with TGFB2 mRNA explained greater than 10% of the variation (R2 = 0.149, p < 0.001) (Figure S6). These observations suggested that stratification of patients according to TGFB2 and marker gene expression would result in relatively independent cohorts of four patient groups for OS comparisons.” (Lines 299-304)

  1. Conclusions” is too long. Try to make conclusion in a paragraph briefly and precisely.

We have taken this suggestion and now appreciate the improved readability of the conclusions.

“This study analyzed the methylation of TGFB2, TGFB3, and IFI27 genes using single-gene markers and median cut-off values for Kaplan-Meier plots. Modest improvements in median overall survival (OS) were noted: 5.7 months for TGFB2 (p = 0.044), 5.2 months for IFI27 (p = 0.036), and 3.7 months for TGFB3 (p = 0.028) methylation. Conversely, high TGFB1 methylation led to a reduced median OS of 4.7 months (p = 0.016). Multivariate Cox proportional hazards analysis confirmed that the prognostic significance of TGFB1, TGFB2, TGFB3, and IFI27 methylations was independent. High TGFB2 methylation was notably associated with improved OS in pancreatic ductal adenocarcinoma (PDAC) patients, showing a hazard ratio (HR) of 0.53 (95% CI: 0.334-0.843; p = 0.007). Survival benefits from TGFB2 methylation increased at the low expression of CD8+ T cell-associated markers. Significant OS improvements can be achieved at low expression levels for CD3D (54.2 months, p < 0.0001), LCK (54 months, p = 0.0009), HLA-DRA (54.9 months, p = 0.0001), and RAC2 (9 months, p = 0.0057) comparing high versus low levels of TGFB2 methylation. Therefore, TGFB2 methylation is a key prognostic marker for PDAC, particularly in immunosuppressed microenvironments with low CD8+ T-cell infiltration. This correlation suggests that targeting TGFB2 mRNA with knockdown strategies, such as OT-101, could enhance PDAC prognosis. Monitoring DNA methylation of TGFB1, TGFB2, and TGFB3 provides a robust method for characterizing tumor microenvironments and selecting candidates for immunotherapy in cold tumors.” (Lines 774-792)

6.The analyses only depend on TCGA PDAC data. Their own PDAC cohort or datasets in GEO should be included to confirm the findings from TCGA database.

We have now included analysis from the independent dataset obtained from a commercial license to 86 PDAC patients that evaluated the correlation of gene expression with OS outcomes.

“4.8. Confirmation of the Prognostic impact of Marker Genes identified from the TCGA dataset

Three out of the four prognostic markers identified from the TCGA dataset (CD3D, LCK and RAC2) were available for evaluation in the Kaplan-Meier Plotter web portal made accessible via a commercial license (https://kmplot.com/private/index.php?cancer=pancreas_rma&p=service, on 26 May 2025 ).  The public version of the portal can be accessed via https://server2.kmplot.com/pancreas (accessed on 26 May 2025 ). This Pancreatic cancer database was compiled from publicly available GEO data and further processed for survival analysis using the web portal. [72,73]”

“The marked improvement of OS outcomes at low levels of TGFB2 and low levels of marker gene mRNA expression compared to high levels of TGFB2 mRNA expression in the TCGA dataset were confirmed for three of these genes (CD3D, LCK and RAC2) evaluable from the independent Kaplan-Meier plotter dataset (Figure 6). For the three marker genes, low levels of marker gene mRNA expression and low levels of TGFB2 mRNA both exhibited improved OS outcomes compared to high levels of TGFB2 mRNA (Figure 6A, C, E; HR ranged from 2.41 to 3.2 when comparing high versus low levels of TGFB2; mOS ranged from 12.9 to 13.2 months for patients expressingTGFB2high mRNA, improving to mOS times ranging between 21.5 to 35.9 months) confirming the results obtained from the TCGA data set. There was no impact of TGFB2 mRNA at high levels of marker gene expression (Figure 6B, D, F) “ (Lines  354-63)

Reviewer 2 Report

Comments and Suggestions for Authors

The manuscript by Trieu et al. analyzed DNA methylation data of 178 PDAC cases from TCGA database. They characterized the impact of TGFB1/2/3 and IFI27 gene methylation on the overall survival in PDAC patients. They also confirmed the prognostic positive impact of low expression of CD8+ T-cell infiltration related marker genes including CD3D, LCK, HLA-DRA, and RAC2, high levels of TGFB2 methylation and low levels of TGFB2 mRNA in PDAC. In summary, this is an interesting study which established a DNA methylation-based prognostic model for PDAC. The comments are listed below.

Major comments:

  1. For justifying the clinical practicality of the model, validation of the TGFB1/2/3 and IFI27 gene methylation prognostic model in a non-TCGA cohort will be important. I suggest the authors to analyze another cohort for verifying the prognostic impact of this model.
  2. Recently, TGFB2 has been shown to confer gemcitabine resistance through upregulation of lipogenesis regulator sterol regulatory element binding factor 1 (SREBF1) (PMID: 38914663). It will be nice to include SREBF1 gene expression in their TGFB1/2/3 and IFI27 gene methylation analysis model.
  3. TGFB signaling includes Smad-dependent and Smad-independent (PI3K-Akt, Ras-Erk, p38, JNK, and GTPases) pathway. It will be interesting to know whether Smad4 gene expression changes in their TGFB1/2/3 and IFI27 gene methylation analysis model.

Minor comments:

Please provide the number of patients in each subgroup of the Kaplan-Meier survival curve.

Author Response

1.For justifying the clinical practicality of the model, validation of the TGFB1/2/3 and IFI27 gene methylation prognostic model in a non-TCGA cohort will be important. I suggest the authors to analyze another cohort for verifying the prognostic impact of this model.

We have now included analysis from an independent dataset obtained under a commercial license, which evaluated the correlation of gene expression with overall survival (OS) outcomes in 86 PDAC patients.

“4.8. Confirmation of the Prognostic impact of Marker Genes identified from the TCGA dataset

Three out of the four prognostic markers identified from the TCGA dataset (CD3D, LCK and RAC2) were available for evaluation in the Kaplan-Meier Plotter web portal made accessible via a commercial license (https://kmplot.com/private/index.php?cancer=pancreas_rma&p=service, on 26 May 2025 ).  The public version of the portal can be accessed via https://server2.kmplot.com/pancreas (accessed on 26 May 2025 ). This Pancreatic cancer database was compiled from publicly available GEO data and further processed for survival analysis using the web portal. [72,73]”

“The marked improvement of OS outcomes at low levels of TGFB2 and low levels of marker gene mRNA expression compared to high levels of TGFB2 mRNA expression in the TCGA dataset were confirmed for three of these genes (CD3D, LCK and RAC2) evaluable from the independent Kaplan-Meier plotter dataset (Figure 6). For the three marker genes, low levels of marker gene mRNA expression and low levels of TGFB2 mRNA both exhibited improved OS outcomes compared to high levels of TGFB2 mRNA (Figure 6A, C, E; HR ranged from 2.41 to 3.2 when comparing high versus low levels of TGFB2; mOS ranged from 12.9 to 13.2 months for patients expressingTGFB2high mRNA, improving to mOS times ranging between 21.5 to 35.9 months) confirming the results obtained from the TCGA data set. There was no impact of TGFB2 mRNA at high levels of marker gene expression (Figure 6B, D, F) “ (Lines  354-63)

2.Recently, TGFB2 has been shown to confer gemcitabine resistance through upregulation of lipogenesis regulator sterol regulatory element binding factor 1 (SREBF1) (PMID: 38914663). It will be nice to include SREBF1 gene expression in their TGFB1/2/3 and IFI27 gene methylation analysis model.

The updated manuscript now includes supplementary figure 12 with the following text in the main document:

“The inclusion of SREBF1 mRNA (HR (95% CI range) = 0.639 (0.44-0.926); p = 0.018) in the multivariate model resulted in a modest reduction in the positive prognostic impact of TGFB2 gene methylation (HR (95% CI range) = 0.629 (0.39-1.016); p = 0.058) compared to the impact of TGFB2 methylation evaluated in the model without SREBF1 mRNA levels (Figure 2: HR (95% CI range) = 0.53 (0.334–0.843); p = 0.007). The multivariate model that included SREBF1 mRNA controlled for the effects of TGFB1 methylation (HR (95% CI range) = 1.43 (0.92-2.223); p = 0.112), TGFB3 methylation (HR (95% CI range) = 0.629 (0.41-0.963); p = 0.033), IFI27 methylation (HR (95% CI range) = 0.459 (0.291-0.723); p = 0.001), age at diagnosis (HR (95% CI range) = 1.029 (1.008-1.051); p = 0.007), and Sex (HR (95% CI range) = 0.856 (0.566-1.294); p = 0.461) (Figure S12).” (Lines 387-96)

“Recently, TGFB2 has been shown to confer gemcitabine resistance by upregulating the lipogenesis regulator sterol regulatory element binding factor 1 (SREBF1) [40]. The inclusion of SREBF1 mRNA (HR (95% CI range) = 0.639 (0.44-0.926); p = 0.018) in the multivariate model resulted in a modest reduction in the positive prognostic impact of TGFB2 gene methylation (Figure S12: HR = 0.629 ; p = 0.058) compared to the impact of TGFB2 methylation evaluated in the model without SREBF1 mRNA levels (Figure 2: HR = 0.53; p = 0.007). These findings suggest that OT-101 may enhance clinical outcomes in PDAC, especially in tumors with low levels of T-cell infiltration. A clinical study is underway to compare the efficacy and safety of OT-101 combined with FOLFIRINOX to FOLFIRINOX alone in patients with advanced or metastatic pancreatic cancer with no planned treatment with Gemcitabine (NCT06079346).”  (Lines 588-98)

3.TGFB signaling includes Smad-dependent and Smad-independent (PI3K-Akt, Ras-Erk, p38, JNK, and GTPases) pathway. It will be interesting to know whether Smad4 gene expression changes in their TGFB1/2/3 and IFI27 gene methylation analysis model.

The updated manuscript now includes supplementary figure 11 with the following text in the main document:

“We further investigated the effects of SMAD4 (Figure S11) to assess the impact of SMAD-dependency [39], and SREBF1 to assess the impact of Gemcitabine resistance [40] (Figure S12) mRNA expression on the prognostic impact of TGFB2 gene methylation. The favorable prognostic impact of TGFB2 methylation (HR (95% CI range) = 0.513 (0.319-0.825); p = 0.006) was not affected by the inclusion of SMAD4 mRNA (HR (95% CI range) = 0.741 (0.518-1.061); p = 0.102) in the multivariate model. This model controlled for TGFB1 methylation (HR (95% CI range) = 1.653 (1.081-2.528); p = 0.02), TGFB3 methylation (HR (95% CI range) = 0.623 (0.405-0.958); p = 0.031), IFI27 methylation (HR (95% CI range) = 0.525 (0.327-0.843); p = 0.008), age at diagnosis (HR (95% CI range) = 1.028 (1.007-1.05); p = 0.01), and Sex (HR (95% CI range) = 0.838 (0.552-1.273); p = 0.407) (Figure S11). “ (Lines 396-406)

“The multivariate Cox proportional hazards model that included the impact of SMAD4 to implicate SMAD-dependent impact of TGFB ligands [39], showed no effect on the positive prognostic impact of TGFB2 gene methylation (Figure S11), suggesting that the TGFB2 methylation was correlated to SMAD-independent pathways. Investigation of the Reactome pathways and mRNA expression of genes negatively correlated with TGFB2 methylation identified three Reactome pathways “ (Lines 413-417)

Minor comments:

Please provide the number of patients in each subgroup of the Kaplan-Meier survival curve.

We have now included the requested information.

Round 2

Reviewer 1 Report

Comments and Suggestions for Authors

I think authors have significantly improved their manuscript. My recommendation is "Accept in present form". 

Reviewer 2 Report

Comments and Suggestions for Authors

In my opinion this manuscript can be published in the current form. Authors have successfully addressed all the criticisms raised.